# A New Analysis of Caldera Unrest through the Integration of Geophysical Data and FEM Modeling: The Long Valley Caldera Case Study

**Fabio Pulvirenti** [1,*], **Francesca Silverii** [2] and **Maurizio Battaglia** [3,4]

1   School of Remote Sensing and Geomatics Engineering, Nanjing University of Information Science and Technology, 219 Ningliu Road, Pukou District, Nanjing 210044, China
2   Department of Physics of Earthquakes and Volcanoes, German Research Centre for Geosciences (GFZ), Helmholtzstraße 6/7, 14467 Potsdam, Germany; silverii@gfz-potsdam.de
3   U.S. Geological Survey, Volcano Disaster Assistance Program, NASA Ames Research Center, Moffett Field, CA 94035, USA; maurizio.battaglia@uniroma1.it
4   Department of Earth Sciences, Sapienza—University of Rome, 00185 Rome, Italy
*   Correspondence: fabiopulvirenti@yahoo.it

**Abstract:** The Long Valley Caldera, located at the eastern edge of the Sierra Nevada range in California, has been in a state of unrest since the late 1970s. Seismic, gravity and geodetic data strongly suggest that the source of unrest is an intrusion beneath the caldera resurgent dome. However, it is not clear yet if the main contribution to the deformation comes from pulses of ascending high-pressure hydrothermal fluids or low viscosity magmatic melts. To characterize the nature of the intrusion, we developed a 3D finite element model which includes topography and crust heterogeneities. We first performed joint numerical inversions of uplift and Electronic Distance Measurement baseline length change data, collected during the period 1985–1999, to infer the deformation-source size, position, and overpressure. Successively, we used this information to refine the source overpressure estimation, compute the gravity potential and infer the intrusion density from the inversion of deformation and gravity data collected in 1982–1998. The deformation source is located beneath the resurgent dome, at a depth of $7.5 \pm 0.5$ km and a volume change of $0.21 \pm 0.04$ km$^3$. We assumed a rhyolite compressibility of $0.026 \pm 0.0011$ GPa$^{-1}$ (volume fraction of water between 0% and 30%) and estimated a reservoir compressibility of $0.147 \pm 0.037$ GPa$^{-1}$. We obtained a density of $1856 \pm 72$ kg/m$^3$. This density is consistent with a rhyolite melt, with 20% to 30% of dissolved hydrothermal fluids.

**Keywords:** numerical modeling; Long Valley Caldera; deformation and gravity joint inversion; topography correction; heterogenous crust; FEM; source parameters; intrusion density

## 1. Introduction

The Long Valley Caldera (LVC), located in east-central California on the western edge of the Basin and Range Province and at the base of the Sierra Nevada frontal fault escarpment, is an east-west elongated oval depression formed by the eruption of the Bishop Tuff, $767,100 \pm 900$ years ago (Figure 1). Beginning in the late 1970s, the caldera entered a period of unrest, without any eruptions, that continues to the present time (e.g., Figure 3 in [1]). The unrest episodes include recurring earthquake swarms beneath the South Moat Seismic Zone (SMSZ) and the Sierra Nevada (SN) block, accelerated inflation of the central Resurgent Dome (RD), variations in the geothermal system and gas emissions around the flanks of Mammoth Mountain (MM) on the southwest margin of the caldera ([1] and references therein).

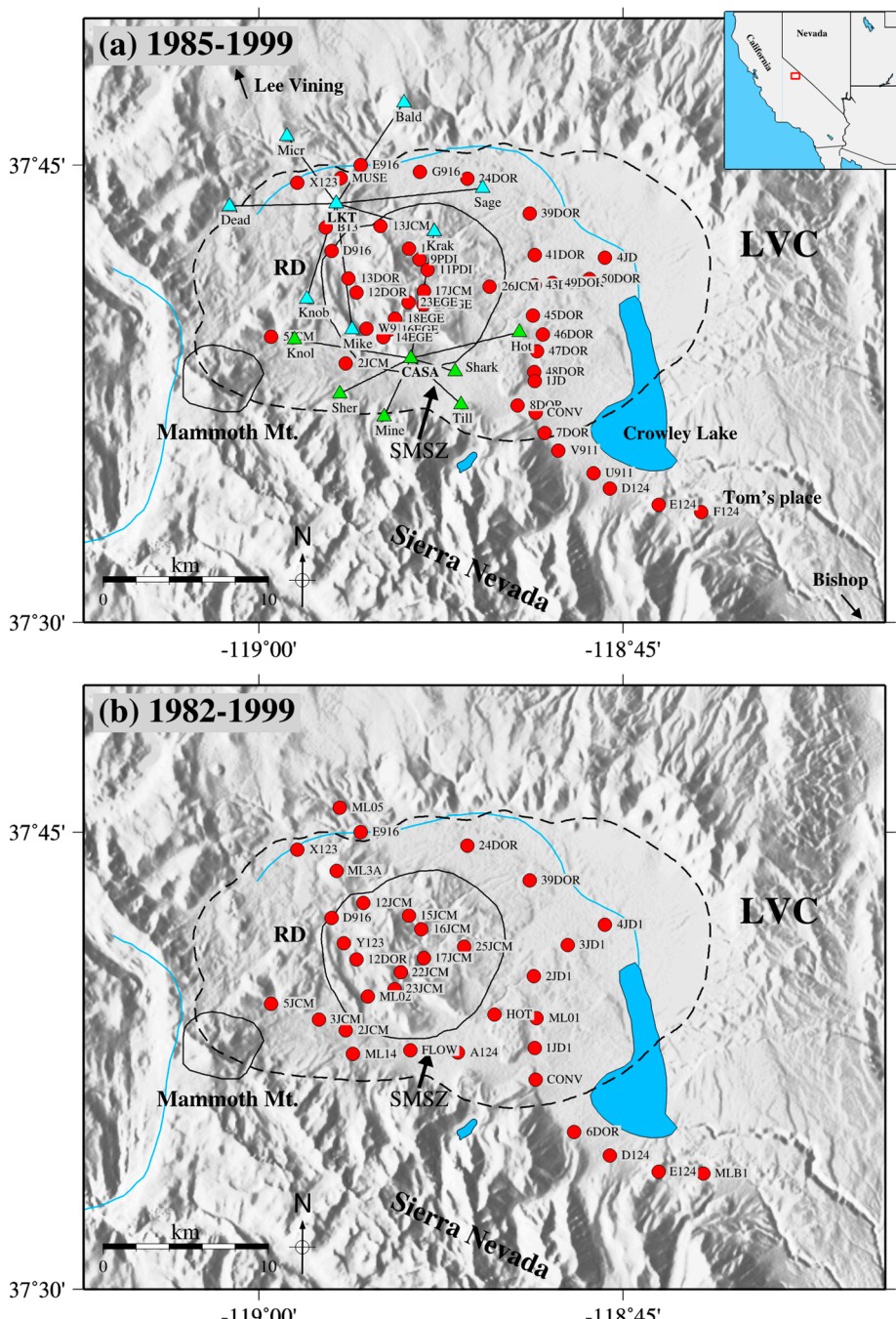

**Figure 1.** Map of Long Valley Caldera (LVC) and geodetic monitoring networks. Solid black lines represent the area of Resurgent Dome (RD) and Mammoth Mountain. Black dashed line outlines the LVC area. (**a**) Sites occupied in 1985–1999. Red circles are leveling stations. Green and cyan triangles refer to EDM baselines referred to common end-points CASA and LKT, respectively. (**b**) Sites occupied in 1982–1999. Red circles are leveling stations. No EDM data are available in this period.

The U.S. Geological Survey (USGS) began an intensive effort to monitor the unrest in LVC between 1975 and 1983 with the setup of new leveling lines in 1982, a two-color Electronic Distance Measurement (EDM) network in 1983, trilateration arrays in 1979, a dense seismic network in 1982, and a high-precision gravity network in 1982. Continuous Global Positioning System (GPS) measurements have been made since 1993. Both ground-based and space geodesy (including satellite interferometry) observations reveal a consistent

radial and upward deformation pattern, centered at the RD and decreasing with radial distance (e.g., [2–8]).

Inferences about the cause of inflation from deformation data indicate that the main inflation source has been relatively stable since the 1980s and consists of a quasi-vertical prolate ellipsoid centered beneath the Resurgent Dome at a depth between 5 and 14 km [4,7–17]. Secondary sources of deformation include a deeper (9–15 km) source beneath the south moat [2,4,11], a small, north-northeast trending dike beneath MM [11,17], and a right-lateral strike-slip motion on west-northwest striking faults in the SMSZ [4].

The processes driving the unrest at LVC remain unclear, with the main likely source of unrest being either a magmatic intrusion into the upper crust [1], or pulses of high-pressure hydrous fluid intrusion into the upper crust [18]. Geologic and petrologic evidence support the hypothesis that the LVC rhyolitic magmatic system is moribund and that the magma body that fed the caldera-forming eruption may now be in the final stages of crystallization. The most recent eruptions along the Inyo Craters/Mono Domes chain and Mammoth Mountain have been fed by a different magmatic system. All of the eruptions inside the LVC have been rhyolitic, with the most recent eruption ~100 ka in the west moat. There has been no eruption on the resurgent dome over the last 500 ka. No significant seismicity and no emission of $CO_2$ or other magmatic gases has been recorded beneath the resurgent dome. Finally, the drilling of the resurgent dome found temperatures of only 100° at a depth of 3 km [18]. On the other hand, several pieces of geophysical evidence point to a possible magma intrusion as the cause of the present unrest. Multiple seismic imaging studies using different techniques (e.g., teleseismic tomography and full-waveform ambient noise tomography) highlighted large low velocity zones in the middle and lower crust, which have been interpreted as evidence of the presence of a partial melt. Different geodetic data (both ground- and satellite-based) measuring deformation at LVC since 1979, have not recorded any substantial deflation episodes yet. This might instead be expected if the inflation involved the injection of hydrothermal fluids with poroelastic swelling followed by diffusion, as observed at other calderas, such as Yellowstone and Campi Flegrei [1].

While ground deformation can provide insights about volume changes in the underground reservoir, it cannot constrain the mass of the intrusions and therefore discriminate between magma and hydrous fluid intrusion. Combined deformation and gravity measurements can be used to infer the density of the intrusive fluids and better define the source of unrest [19–27]. Given the density difference between silicate melts (~2300 kg/m$^3$) and hydrothermal fluids (~800 kg/m$^3$, [28]), density estimates can, in principle, be used to distinguish between these two possible sources of caldera unrest.

Gravity measurements at LVC have been conducted yearly between 1980 and 1985 and repeated in 1998 and 1999 [28]. In this period, the RD experienced a quasi-steady uplift, with accelerated phases in 1989–1990 and 1997–1998, when the most rapid deformation occurred (e.g., [4,13]). These data have been analyzed, together with different kinds of deformation records (EDM, leveling, GPS, InSAR) in different studies using analytical models and considering increasing complexities, from point source to tilted finite ellipsoidal source, from homogeneous to vertically layered elastic half-space [12,28–31]. The results of these studies suggest that gravity data are more compatible with the addition of a magma intrusion than pulses high-pressure hydrous fluids.

In this paper, we consider the 1982–1999 unrest period. This time has the best and most complete gravity dataset. We perform numerical computations based on the finite element method (FEM), exploring the effect of topography and realistic medium heterogeneities on the parameters (e.g., location, depth, density) of the source of unrest. We first invert EDM and leveling data from 1985 to 1999 in order to constrain the location, depth, and geometry of the unrest source. We then use the inferred source to model the deformation and gravity changes between 1982 and 1999, and to compute the source volume change, and density.

In Section 2, we present the methods including data, model setup and model computations; in Section 3, we show the results; in Section 4, we discuss our findings and conclusions.

## 2. Methods

### 2.1. Data

We used the data from the Long Valley Caldera GIS Database ([13]; https://doi.org/10.3133/ds81, accessed on 15 February 2021). The database includes extensive geologic, monitoring, and topographic datasets from studies conducted in Long Valley caldera between 1975 and 2001. The unrest is investigated using three sets of data: baseline length changes (an approximation of horizontal deformation) from two-color EDM, vertical deformation from a combination of GPS and leveling, and gravity changes.

The two-color EDM network consists of two sets of seven baselines. The first set is formed by the sites Hot, Knol, Krak, Mine, Shark, Sher and Till, observed from the central monument CASA (green triangles in Figure 1a). The second set is formed by the sites Bald, Dead, Knob, Krak, Micr, Mike and Sage, observed from the LKT monument (cyan triangles in Figure 1a). Measurements at these baselines span the 1985–1999 inflation period, which is included in the targeted time in this work (Supplementary Materials, Table S1). The methods used to extract the displacement and its error for each of the baselines are described in [11,17]. The EDM deformation data that were used are from [13].

Vertical deformation (uplift) measurements were taken during different leveling surveys along the 65-km-long line along Hwy 365 from Tom's Place to Lee Vining, and along several other routes within the caldera, and are obtained by combining leveling and GPS data (Figure 1a). Complete leveling of the caldera occurred each summer from 1982 to 1986, and in 1988 and 1992. In 1999, reference [13] occupied 44 leveling benchmarks with GPS to bring up to date the direct measurement of vertical deformation. The data sets employed in this work consist of the 44 benchmarks with leveling and GPS for the period 1985–1999 [13], and 34 benchmarks with leveling and GPS for the period 1982–1999 [28] (red circles respectively in Figure 1a,b; Supplementary Materials, Tables S2 and S3). The benchmark C916, located near Lee Vining (Mono Lake), is the elevation datum for the vertical deformation. The standard error for each elevation difference is calculated according to [13].

The Long Valley caldera gravity monitoring network is centered near Tom's Place (the primary reference station) and extends from the Sierra Nevada west of Lee Vining, CA, southeastward to a station in the White Mountains east of Bishop, CA [32]. Gravity data (gravity changes, noise from the water table and gravity corrected for the water table and free-air effect) are from [28], Supplementary Materials, Table S4. In Section 2.4, we employ the gravity data corrected for water table and free air contribution to estimate the density of the intrusion.

### 2.2. Model Setup

We develop a three-dimensional (3D) numerical model using the finite element method (FEM) and the software COMSOL Multiphysics (www.comsol.com, accessed on 15 February 2021). The geometry refers to a Cartesian reference system and is composed of a domain of 120 km × 130 km. The model is 40 km deep (up to the Moho depth in the area [33,34]), with zero depth corresponding to the sea level. The chosen size represents a crust portion which includes the LVC and a significant part of its surroundings (Figure 2a).

Inside the domain, we assume the existence of an internally pressurized ellipsoidal prolate cavity that we invert for its location, dimensions, and overpressure (Figure 2b; see Section 2.3).

Pressurized cavities of simple geometry can mimic/approximate the crustal stress field produced by the actual source. None of these geometries reproduced an actual source. The actual deformation source, beneath the resurgent dome, is probably a network of fractures filled with fluids (or magma) ascending from the crystallizing Pleistocene pluton below [18].

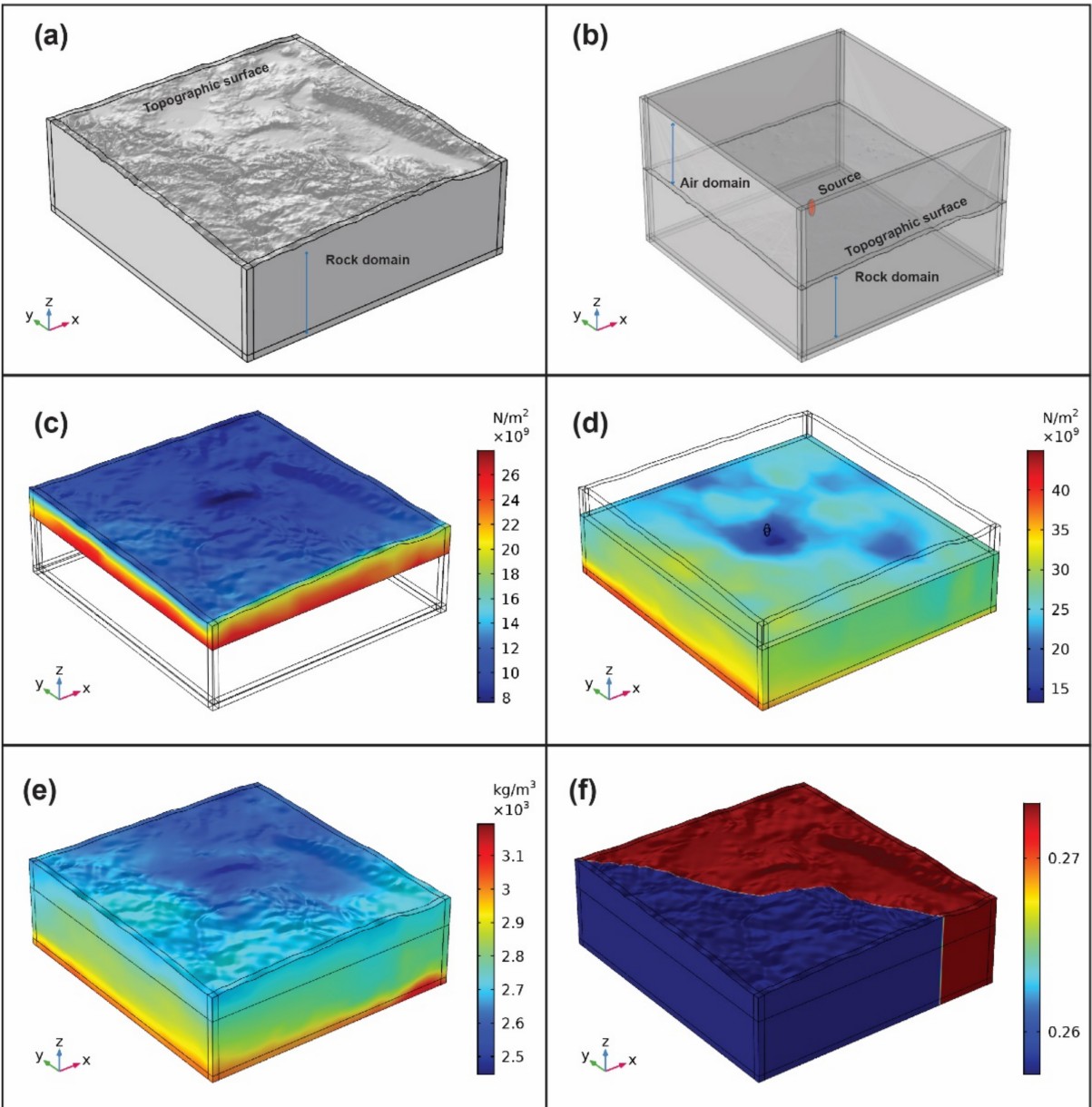

**Figure 2.** Model geometry representing (**a**) the rock domain with topography, (**b**) the rock + air domains in transparency with the ellipsoidal source beneath the resurgent dome, (**c**) quasi-static bulk modulus, (**d**) dynamic bulk modulus, (**e**) density, (**f**) Poisson's ratio.

We explore three different crust configurations. A homogeneous elastic domain with flat stress-free top surface (labeled HF) representing the average altitude of the area (~2300 m a.s.l.), a homogeneous elastic domain with topography (labeled HT), and a fully heterogeneous elastic domain with topography (labeled HeT). The topographic surface is generated by using the STRM digital elevation model (DEM) from the USGS Earth Explorer [35], resampled at 600 m resolution. Material heterogeneities (density, bulk modulus, shear modulus and Poisson's ratio; Figure 2c–f) are obtained from pressure ($V_P$) and shear ($V_S$) wave velocity distributions. Shear wave velocities $V_S$ are from [36] while pressure velocities $V_P$ are calculated from $V_S$ using a $V_P/V_S$ ratio of 1.75 for the SN block [37] and of 1.79 elsewhere, with a gradient $d(V_P/V_S)$ of 3% [33,38]. $V_P$, $V_S$ velocities

are converted into Poisson's ratio ($v$), density ($\rho$) and dynamic Young's Modulus ($E$) using the equations in [39]:

$$v = 0.5 \times \left[\left(\frac{V_P}{V_S}\right)^2 - 2\right] \bigg/ \left[\left(\frac{V_P}{V_S}\right)^2 - 1\right] \tag{1}$$

$$\rho = 1.6612V_P - 0.4721V_P^2 + 0.067V_P^3 - 0.0043V_P^4 + 0.000106V_P^5 \tag{2}$$

$$E = \frac{V_P^2\rho(1+v)(1-2v)}{(1-v)} \tag{3}$$

However, to properly represent the medium strain rate in a quasi-static condition, we need to refer to quasi-static mechanical properties. Laboratory tests [40,41] show, in fact, that for lithostatic pressures in the range 1–3 kbar (3.8 to 11.5 km depth), the ratio between quasi-static and dynamic bulk modulus $K_s/K_d$ for granite is different from 1 and can vary between 0.5 (at 0.09 kbar–0.4 km depth) to 0.9 (at 3 kbar–11.5 km depth). For the range 0–3 kbar of lithostatic pressure (equivalent to the distance between the top surface and 11.5 km depth), we calculate the dynamic bulk modulus from $V_P$, $V_S$ values and multiply it by the $K_s/K_d$ ratio values from [40] at the corresponding lithostatic pressure (depth) level to estimate the quasi-static bulk modulus. The relation between quasi-static and dynamic mechanical properties is empirical and depends on several factors including stress state and stress history [40,41], however our approach leads to a better characterization of the material response with respect to what can be obtained using pure dynamical properties. An interpolation function guarantees a smooth transition between different lithostatic pressure levels. At a greater depth, where the lithostatic pressure is higher than 3 kbar, we assume a $K_s/K_d$ ratio of 1. From the quasi-static bulk modulus, we can calculate the quasi-static shear modulus. Since Poisson's ratio is not expected to change significantly [42], we can retain its dynamic value. Crust properties are summarized in Table 1 and represented in Figure 2c–f.

**Table 1.** Material property parameters used for the models.

| Material Parameter | Homogeneous Rock Domain | Heterogeneous Rock Domain | Air Domain [1] |
|---|---|---|---|
| Young's Modulus [GPa] | 45 | 10–60 | - |
| Bulk Modulus | 31 | 8–45 | |
| Density [kg/m$^3$] | 2800 | 2450–3200 | 1 |
| Poisson's ratio | 0.26 | 0.25–0.27 | - |
| Shear modulus | 18 | 4–24 | - |
| $\beta_c$ [2] [GPa$^{-1}$] | 0.049 | see Section 2.4 | - |

[1] Air domain fluid characteristics are not solved. [2] Compressibility of the reservoir due to medium elasticity and reservoir shape; for A $\cong$ 3, $\beta_c = 7/8\mu$ (see Section 2.4).

In terms of boundary conditions, the model bottom is fixed, the top surface is stress-free while at the lateral boundaries we apply a roller condition (no displacement in the direction normal to the boundary). An infinite element condition, set at the lateral and bottom boundaries, simulates the far-field, and guarantees that the displacement vanishes at a very far distance from the original geometric size, thus avoiding any boundary effects. We prescribe a parametrized overpressure on the boundaries of the ellipsoidal cavity. The model domain is meshed with tetrahedral elements while the source boundaries and the top surfaces are discretized with triangular elements. Automatic adaptive mesh refinement tests are carried until an optimal performance is found without further variation of the output. The mesh for the whole domain is shown in Figure 3.

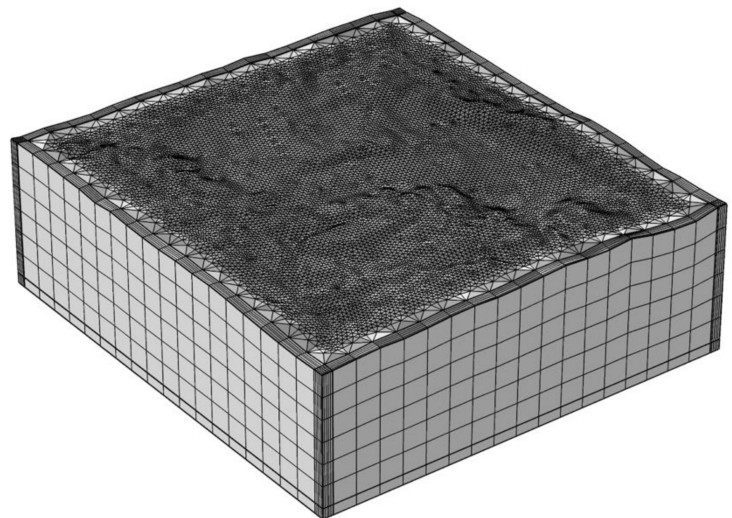

**Figure 3.** Domain mesh (example with rock domain and topography). The top surface is refined to accurately capture the topographic relief and the deformation pattern generated by the source overpressure.

We validate the FEM model for the deformation by performing a benchmark calculation for a vertical prolate ellipsoid in a flat homogeneous domain and comparing the numerical results against the analytical solution by [43], see Appendix A.

### 2.3. Inverse Modeling of Deformation Data

We use the FEM model described in Section 2.2 to perform numerical inversions of EDM baseline changes and uplift (leveling-GPS). Inversions are performed in two steps.

In the first step, we jointly invert the EDM and leveling-GPS data for the period 1985–1999 [13] and for each model configuration (with/without topography or with topography and heterogeneities) we infer the best-fit source dimensions, position, and overpressure. In the second step, we keep the deformation source stable in size and location and further optimize only for the source overpressure by performing a second inversion of the leveling-GPS data for the period 1982–1999 [28]. This second step provides the source parameters needed to model the gravity changes.

Inverse modeling is performed using the Nelder-Mead solver [44] and coupling the structural mechanics and the optimization module in COMSOL Multiphysics. The link between the data to invert and the source parameters is built by setting up the objective function [45]

$$F_i = [(M_i - D_i) \times W_i]^2 \tag{4}$$

where

$$W_i = a_i / \sum_j a_j \tag{5}$$

and

$$a_i = \left| \frac{D_i}{E_i} \right| \tag{6}$$

$M_i$ are the modeled data, $D_i$ the observed data, $E_i$ the observation error, $W_i$ the weights and the index $i$ relates to each benchmark. The inversion goal is to minimize the least weighted squares Equation (4).

The inversion of EDM and leveling-GPS datasets for the period 1985–1999, is made considering the following seven parameters of the ellipsoidal source: the semiaxes ($E_a$, $E_b$ and $E_c$), oriented along the cartesian $x$, $y$ and $z$ axis respectively; the horizontal position along the $x$ and $y$ direction ($E_x$, $E_y$) with respect to a reference point located at longitude $-119°$W, latitude $37.5°$N; the source vertical position ($E_z$), with $E_z = 0$ m at the sea level; the overpressure ($\Delta P$) applied at the source internal boundaries. The first three parameters control the source geometry, the second three control the source position and the last one

controls the source overpressure. The source is assumed to be vertical, with a plunge of 90°. For each parameter we assign an initial value (used in the first iteration), and lower and upper bounds to search for reasonable values during the computation (Table 2). Initial values and ranges of parameters are based on results from previous studies (e.g., reference [7] and references therein). In particular, because of a poor data coverage in the south caldera rim, we constrained the north-south source position $E_y$ to fall within the resurgent dome area. The solver is set to perform a maximum number of 400 model evaluations. This threshold has been chosen by looking at the convergence rate and considering that further evaluations no longer have any significant impact on the value of the objective function.

**Table 2.** Initial values and ranges of the parameters used in the FEM model inversions.

| Parameter Name | Initial Value [m] | Lower Bound [m] | Upper Bound [m] |
|---|---|---|---|
| $E_a$ (*x*-semiaxis) | 1500 | 500 | 5000 |
| $E_b$ (*y*-semiaxis) | 1500 | 500 | 5000 |
| $E_c$ (*z*-semiaxis) | 3000 | 500 | 5000 |
| $E_x$ (center *x*-coord) [1] | 8000 | 4000 | 12,000 |
| $E_y$ (center *y*-coord) [1] | 20,400 | 18,000 | 23,000 |
| $E_z$ (depth) [2] | 5000 | 4000 | 8000 |
| **Parameter Name** | **Initial Value [Pa]** | **Lower Bound [Pa]** | **Upper Bound [Pa]** |
| $\Delta P$ (overpressure) | $7.00 \times 10^7$ | $5.00 \times 10^7$ | $1.00 \times 10^8$ |

[1] $E_x$ and $E_y$ represent the source center coordinates along the x and y direction with respect to a reference point located at longitude $-119°$W and latitude $37.5°$N. [2] Source depth relative to the stress-free surface and not the sea level, i.e., accounting for the average elevation of LVC area (~2300 m a.s.l.).

According to [29], the inflation source could be slightly tilted with a dip angle between 91 and 105 degrees. To check whether this was the case for our source, we performed preliminary tests, including additional parameters for a source rotation of $\pm10$ degrees around each of the three cartesian axes. Results showed that in all cases (with/without topography or heterogeneities) the optimal rotation is minimal, $<1°$ around *x* and *y* axis and $<2°$ degrees around the *z* axis. For this reason, we did not include these parameters in our inversions.

### 2.4. Computation of Gravity Changes

The total gravity change recorded at a benchmark during unrest episodes contains the effect of different contributions: (i) the free-air effect, due to the vertical displacement of the benchmarks at the ground surface during unrest; (ii) the water table effect, proportional to the water table level change in the area; (iii) the deformation effect, due to the coupling between elastic deformation and gravity; and (iv) the residual gravity, which depends on the density change related to the introduction of the new mass into the pressurized volume (e.g., [20–22]). Furthermore, the estimation of gravity variation is sensitive to model complexities, such as volumetric source geometry, topography, material heterogeneities and fluid compressibility (e.g., [22–27]).

The best fit parameters of the ellipsoidal source from the three different crust configurations (HF, HT, HeT; Figure 2), are used to compute the gravity change at the free surface. Following the methodology in [46], we first compute the displacement field from the previously estimated best source models (cf. Section 2.3), and we then solve the Poisson's equation relating the gravity potential ($\varphi_g$) to the change in density distribution ($\Delta\rho$) caused by the subsurface mass redistribution $\nabla^2\varphi_g = -4\pi G\Delta\rho(x, y, z)$, where $G$ is the gravitational constant. The gravity change can be then computed as $\delta_g = -\partial\,\varphi_g/\partial\,z$. In particular, the relation between the gravity potential $\varphi_g$ and the density variations can be expressed by the following contributions:

$$\nabla^2\varphi_{g1} = 4\pi G(\boldsymbol{u}\cdot\nabla\rho_0) \qquad (7)$$

$$\nabla^2 \varphi_{g2\Delta V} = 4\pi G(\boldsymbol{u} \cdot \nabla \rho_0) \tag{8}$$

$$\nabla^2 \varphi_{g3} = 4\pi G(\rho_0 \nabla \cdot \boldsymbol{u}) \tag{9}$$

$$\nabla^2 \varphi_{g2V} = 4\pi G(\rho_{in}) \tag{10}$$

where $\boldsymbol{u}$ is the inflation-related displacement field, $\rho_0$ the embedding medium density and $\rho_{in}$ is the density of the intruding fluid. Equation (7) gives the gravity contribution $\delta_{g1}$ due to the displacement of density boundaries in heterogeneous media, corresponding to the Bouguer correction at the surface in case of flat homogeneous models. Equation (8) gives the gravity contribution $\delta_{g2\Delta V}$ due the displacement of the source boundaries, which implies replacement of the surrounding mass. Equation (9) gives the term $\delta_{g3}$, which considers the effect of dilatational/compressional strains in the host rock, while Equation (10) gives the term $\delta_{g2V}$ which considers the input of material (of density $\rho_{in}$) into the source volume [21]. Equations (7)–(9) can be used to compute the massless deformation contribution to the gravity changes while Equation (10) represents the contribution due to the source mass change.

To numerically solve the Poisson's equations, we modify the model geometry by adding an additional domain with same size of the rock domain (Figure 2b), but made of air (assuming $E = 1$ Pa, $\rho = 1$ kg/m$^3$, $\nu = 0.25$; Table 1). Furthermore, solving for all contributions to the gravity potential requires the embedded source to be a domain and not a cavity, as done during the inversion of displacements. Poisson's Equations (7) and (8) are solved on the stress-free surface and on the source boundaries, respectively. Poisson's Equation (9) is solved on the domains surrounding the source, and (10) is solved on the source domain.

We validate the FEM model by performing a benchmark calculation for a vertical prolate ellipsoid in a flat homogeneous domain and comparing the numerical results to the analytical solutions by [47] (see Appendix A).

When estimating the gravity changes due to reservoir inflation, it is important to consider that the volume change accommodating for the input of new mass could arise, not only from the expansion of the source wall that deforms the surrounding medium, but also from the compression of the material stored in the reservoir (e.g., [25,48–50]). The relation between the actual volume of the mass intrusion, $\Delta V_m$, and the volume change from the inversion of deformation data (geodetic volume, cf. Section 3.2), $\Delta V$, is (e.g., [48,51])

$$\Delta V_m = \Delta V \times \left(1 + \frac{\beta_m}{\beta_c}\right) = \Delta V \times r_V \tag{11}$$

where $\beta_m = \frac{1}{K_m}$ is the compressibility of the material stored in the reservoir, $\beta_c = \frac{1}{K_c}$ is the compressibility of the reservoir due to medium elasticity and reservoir shape, $K_m$ and $K_c$ are the bulk moduli, $r_V$ is the volume ratio, and $\beta_m$ is a function of several parameters, like pressure, gas volume fraction, temperature, phenocryst content and source depth (e.g., Table 3 from [52]). Finite element calculations of reservoir compressibility $\beta_c$ as a function of the source geometric aspect ratio $A = \frac{E_c}{\sqrt{E_a E_b}}$ indicates that in our case $\beta_c \approx \frac{7}{8\mu}$, where $\mu$ is the shear modulus (see Figure 5 in [53], and Tables 1 and 3). $\beta_c$ can also be computed as [48]:

$$\beta_c = \left(\frac{1}{V}\right)\left(\frac{\Delta V}{\Delta P}\right) \tag{12}$$

where $\Delta P$ is the overpressure, and $V$ is the source volume before the application of the overpressure (Table 4). Finally, the density corrected for the effect of compressibility ($\rho_{cmp}$) can be computed as:

$$\rho_{cmp} = \rho_{in}\left(\frac{1}{r_V}\right) \tag{13}$$

**Table 3.** Best fit source parameters, and associated uncertainties estimates σ, obtained from the joint inversion of EDM + Leveling data for the period 1985–1999 and optimal overpressure obtained from the inversion of leveling data for the period 1982–1999. HF: homogeneous flat crust; HT: homogeneous crust with topography; HeT: heterogeneous crust with topography.

| Model | $E_a$ (m) | σ | $E_b$ (m) | σ | $E_c$ (m) | σ | A | σ | $E_x$ (m) | σ | $E_y$ (m) | σ | $E_z$[1] (m) | σ | ΔP 1985–1999 (MPa) | σ | ΔP 1982–1999 (MPa) | σ |
|---|---|---|---|---|---|---|---|---|---|---|---|---|---|---|---|---|---|---|
| HF | 1726 | 149 | 1491 | 141 | 4553 | 605 | 2.8 | 0.8 | 9145 | 374 | 18,006 | 921 | 7674 | 634 | 69 | 2 | 88 | 2 |
| HT | 1865 | 162 | 1556 | 140 | 4136 | 419 | 2.4 | 0.6 | 9077 | 397 | 18,009 | 919 | 7610 | 628 | 67 | 2 | 85 | 2 |
| HeT | 1217 | 146 | 1133 | 150 | 3032 | 254 | 2.6 | 0.6 | 8958 | 487 | 18,000 | 1039 | 7519 | 521 | 65 | 2 | 84 | 2 |

[1] Note that here we indicate the source depth with respect to the stress-free surface and not the sea level, i.e., accounting for the average elevation of LVC area (~2300 m a.s.l.).

**Table 4.** Density values of the intrusion, and associated uncertainties estimates σ, obtained from the inversion of residual gravity for 1982–1999.

| Model | $V$ [km³] | σ | $\Delta V$ [km³] | σ | $\Delta P$ [MPa] | σ | $\beta_c$ [GPa$^{-1}$] (*) | σ | $\beta_m$ [GPa$^{-1}$] | σ | $r_V$ | σ | $\rho_{in}$ [kg/m³] | $\rho_{cmp}$ [kg/m³] | σ |
|---|---|---|---|---|---|---|---|---|---|---|---|---|---|---|---|
| HF | 49 | 9 | 0.21 | 0.04 | 88 | 2 | 0.049 | 0.014 | 0.026 | 0.001 | 1.53 | 0.15 | 2670 | 1741 | 172 |
| HT | 50 | 8 | 0.21 | 0.04 | 85 | 2 | 0.049 | 0.012 | 0.026 | 0.001 | 1.53 | 0.13 | 2720 | 1782 | 151 |
| HeT | 17 | 3 | 0.21 | 0.04 | 84 | 2 | 0.147 | 0.037 | 0.026 | 0.001 | 1.18 | 0.05 | 2184 | 1856 | 72 |

(*) Equation (12).

## 3. Results

### 3.1. Deformation: Best Fit Source

We find that the three different crust configurations (homogeneous, flat elastic half-space HF; homogeneous elastic domain with topography, HT; heterogeneous elastic domain with topography, HeT) give similar results for the position ($E_x$, $E_y$), depth ($E_z$), geometric aspect ratio (A), and pressure change ($\Delta P$) of the source (Table 3).

The nonlinearity of the inverse problem makes the evaluation of uncertainties difficult; nonlinear error propagation is a difficult problem to address, COMSOL does not have a feature that allows extraction of the covariance matrix, and the model covariance matrix may not give a good estimate of the uncertainties [54]. A solution could be to employ a Monte Carlo method. Unfortunately, this method requires each model to be run thousands of times. We employ the result from the inversions (350 to 400 runs) to mimic a Monte Carlo method and obtain an estimate of the uncertainties of the source parameters. We then propagate the errors to the density results (see Appendix B).

The source is moved about 1 km eastwards ($E_x$), 2.4 km southwards ($E_y$) and 600 m deeper ($E_z$), with respect to its starting position and starting depth. No major differences can be seen between a homogeneous flat crust (HF), homogeneous crust with topography (HT) or heterogeneous crust with topography (HeT). The source size is similar for the HF and HT crust models, while we can observe in the HeT crust model a significant reduction of about 1/3 of all semiaxes. The source shapes for each crust model before and after inversion are showed in Figure 4. Although the main source parameters are similar for the three different models (Table 3), material heterogeneities make a difference, especially in the estimate of the absolute volume of the source (see Table 4).

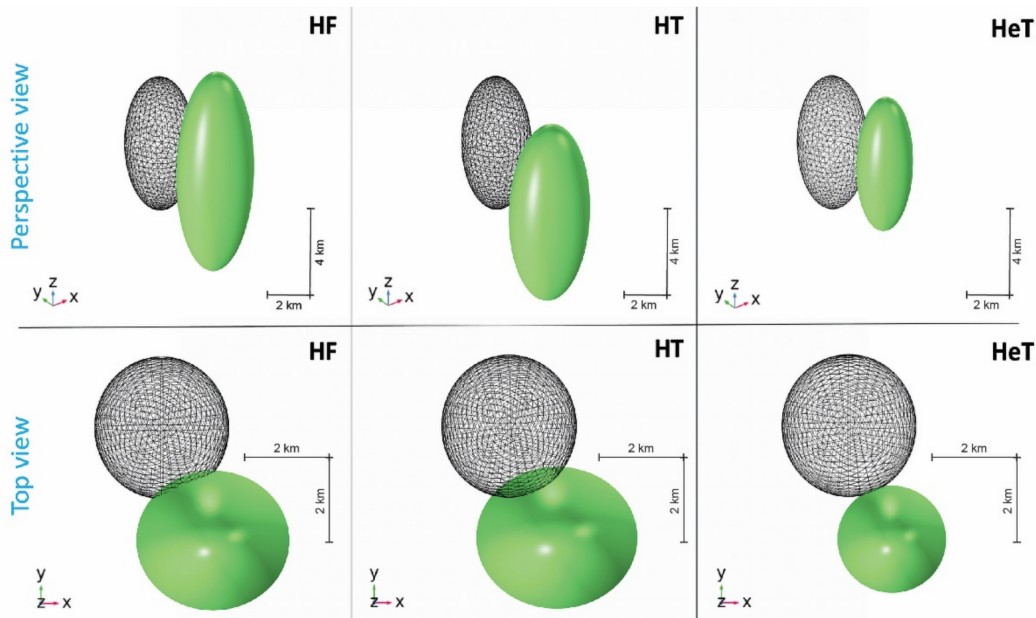

**Figure 4.** Perspective views and top views of the source shape before (black wireframe) and after (green shaded) the joint inversion of EDM and leveling data for the period 1985–1999. HF: homogeneous flat crust; HT: homogeneous crust with topography; HeT: heterogeneous crust with topography.

Figure 5 shows the total displacement (combination of vertical and horizontal displacements) at the free surface for each model configuration. For the HF and HT cases we can clearly observe two lobes with higher displacement northwards and southwards of the source with the topography playing a minor damping role and a slight clockwise rotation of the northern lobe. When heterogeneities are introduced (HeT), the southern lobe disappears while the northern lobe further rotates clockwise and shows an area of maximum total displacement.

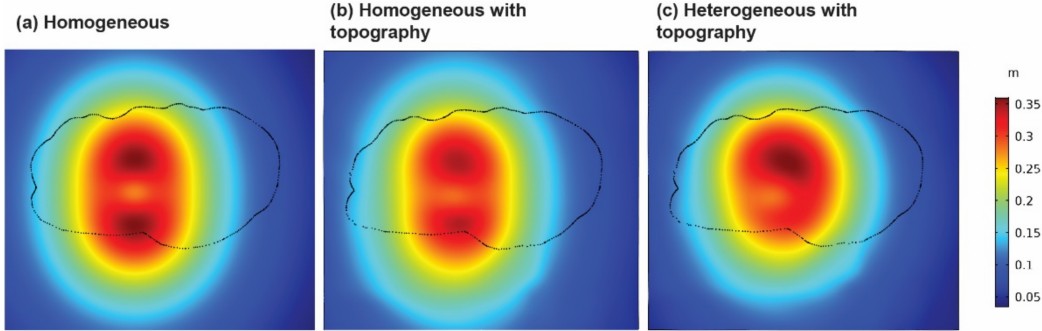

**Figure 5.** Total displacement at the free surface (top view) from the joint inversion of EDM and leveling data for the period 1985–1999. Black dotted line represents the caldera border. (**a**) Homogeneous model without topography. (**b**) Homogeneous model with topography. (**c**) Heterogeneous model with topography. The color scale (0.05–0.35 m) is the same for the three panels.

### 3.2. Fit to Deformation Data

Figures 6 and 7 compare the modeled and observed values for horizontal (EDM baseline length changes) and vertical displacements (uplift) for 1985–1999, obtained from the numerical joint inversions of EDM and leveling data. Observed values and numerical results for the baseline changes and for the leveling data over the period 1985–1999 are reported in Tables S1 and S2 in Supplementary Materials.

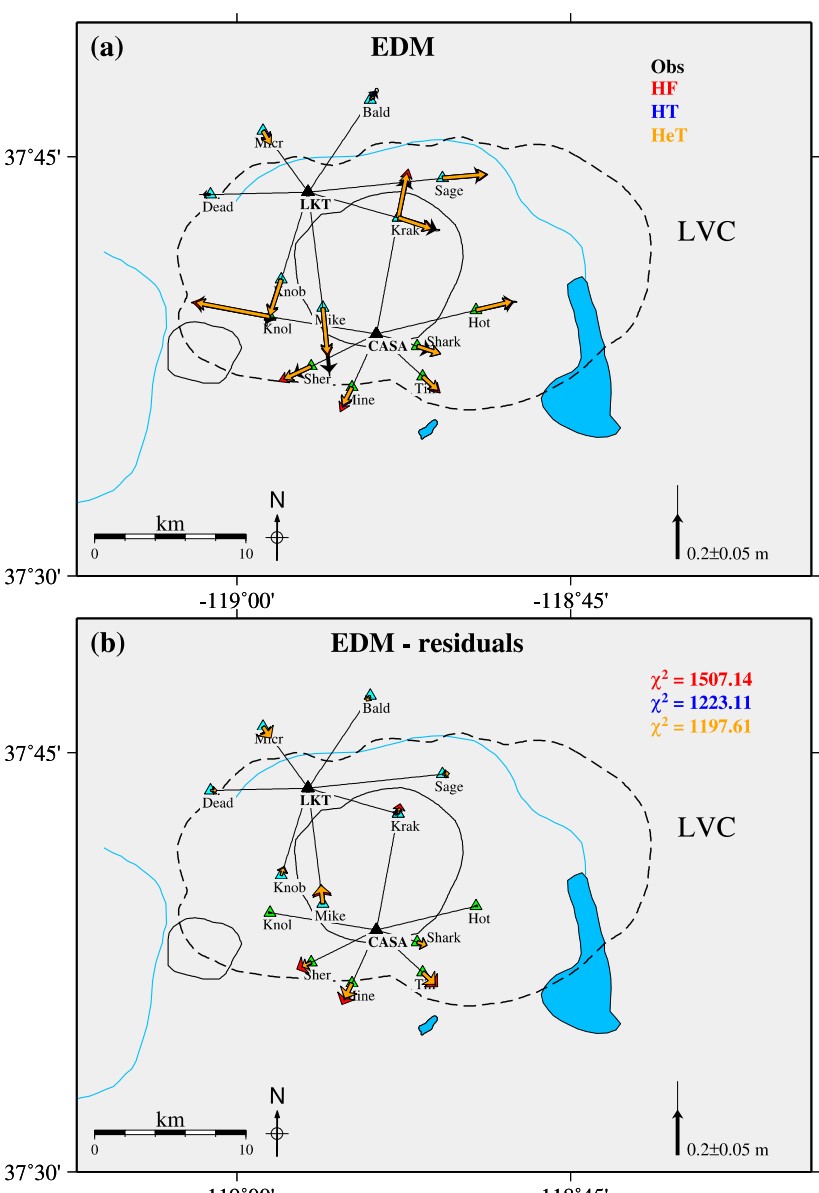

**Figure 6.** (**a**) Comparison between observed (black arrows with thin solid line as error) and modeled (colored arrows corresponding to different model configurations) EDM baseline length changes. (**b**) Corresponding residuals between modeled and observed baseline length changes. Obs: observations; HF: homogeneous flat crust; HT: homogeneous crust with topography; HeT: heterogeneous crust with topography. $\chi^2$ values for each model are shown on top right.

Results for EDM show a good agreement between models and observations (Figure 6a) except for LKT-MIKE baseline which is slightly underestimated in all three crustal models. However, the difference between the models' results and the measurement for LKT-MIKE is within the data uncertainty (thin solid black lines). From the EDM residuals (Figure 6b) we can observe that the fit improves when we add topography ($\chi^2$ decrease by 19% from HF to HT, red and blue arrows) and material heterogeneities ($\chi^2$ decrease by 21% from HF to HeT, red and yellow arrows).

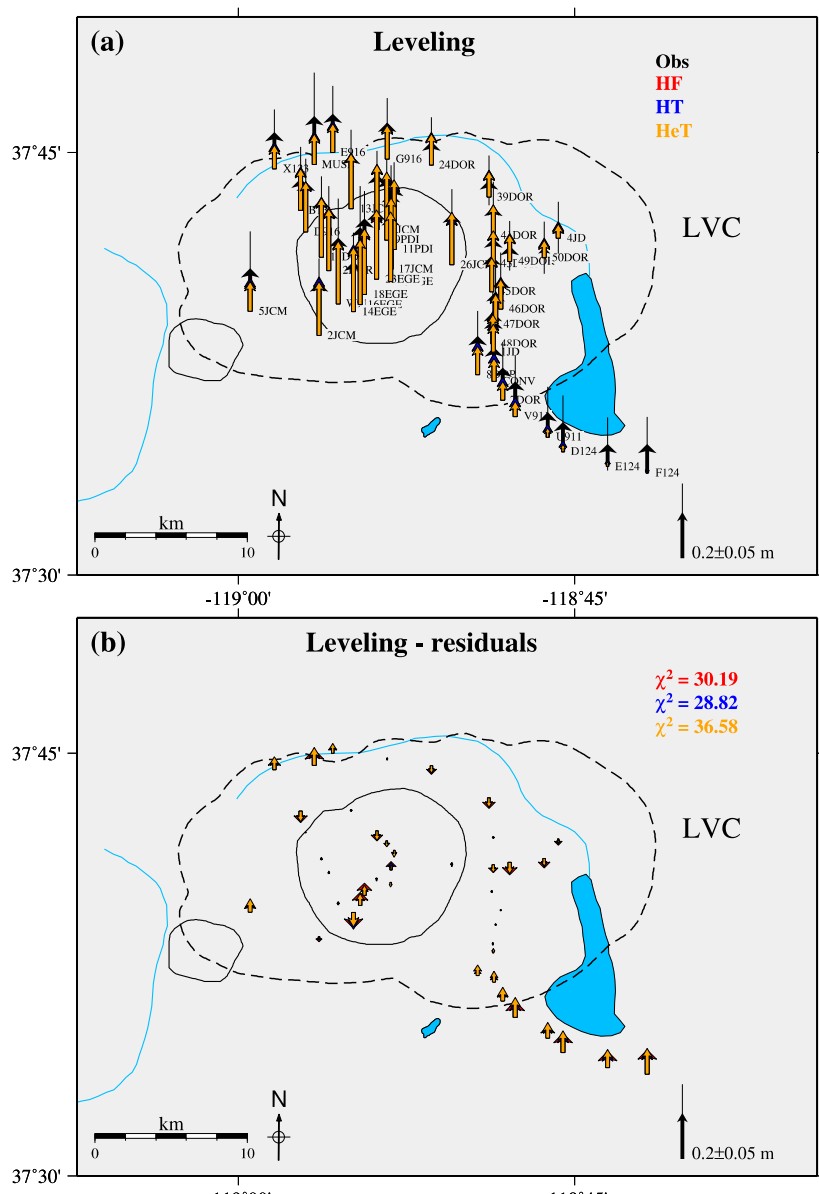

**Figure 7.** (**a**) Comparison between observed (black arrows with thin solid line as error) and modeled (colored arrows corresponding to different model configurations) uplift for the period 1985–1999. (**b**) Corresponding residuals between modeled and observed uplift. Obs: observations; HF: homogeneous flat crust; HT: homogeneous crust with topography; HeT: heterogeneous crust with topography. $\chi^2$ values for each model are shown on top right.

The inversion results for the leveling data (Figure 7a) show a good agreement between the observed and modeled data at the benchmarks located inside the caldera border (black dashed line). However, the model underestimates the observed uplift by 5–10 cm at the benchmarks located outside the southeastern (SE) caldera border (near Crowley Lake) and by 2–5 cm for the benchmarks located at the northwestern (NW) caldera border (Figure 8). This discrepancy influences the $\chi^2$ value (Figure 7b), which slightly decreases when topography is included (5% decrease in $\chi^2$ from HF to HT) but increases when we add the material heterogeneities (16% increase in $\chi^2$ from HF to HeT). This is probably because, in the HeT model, the material outside the caldera area is stiffer than the material inside it (Figure 2).

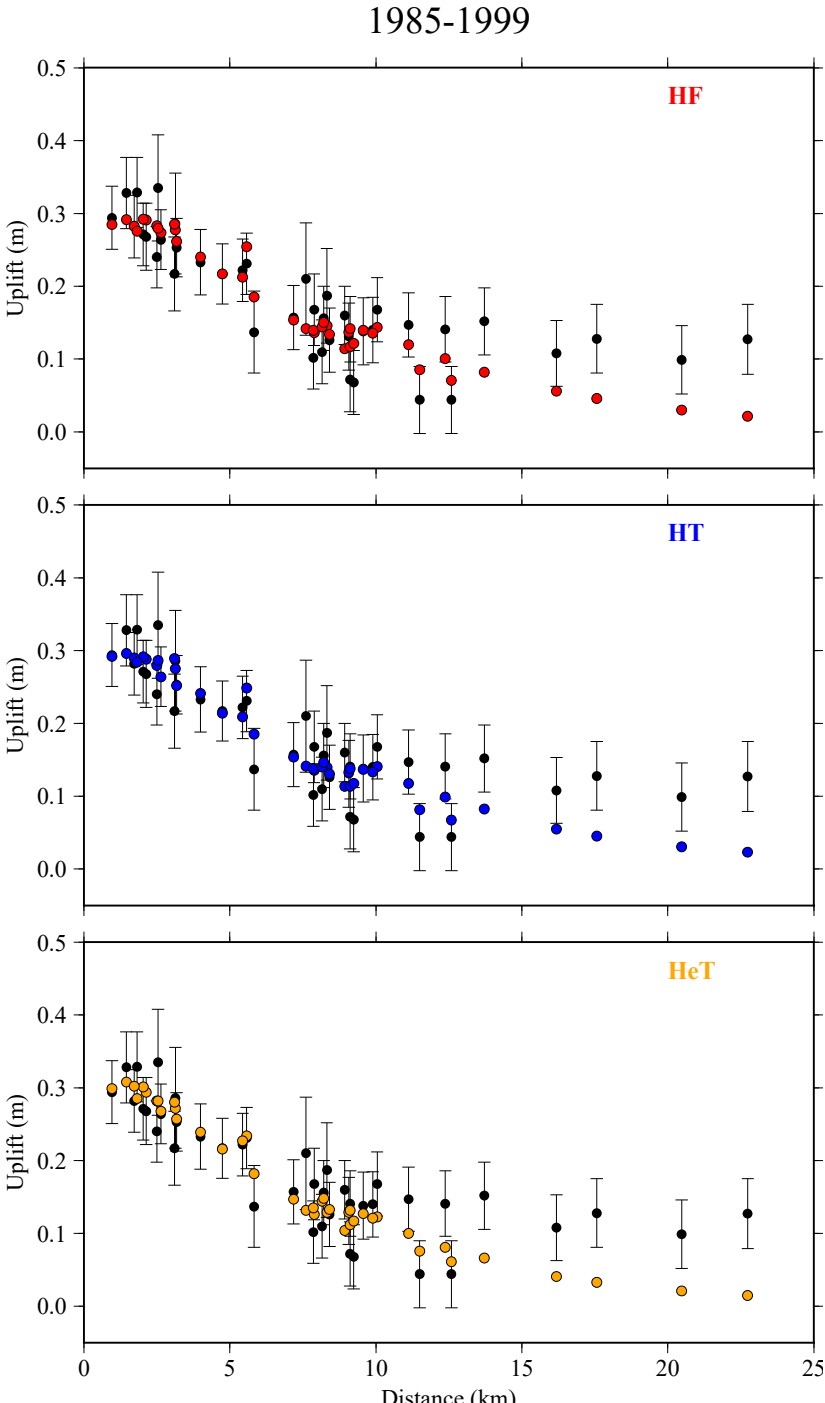

**Figure 8.** Comparison between observed (black circles with 1-sigma error bars) and modeled (colored circles corresponding to different model configurations) uplift at leveling benchmarks for the data 1985–1999 ordered according to the horizontal distance from the resurgent dome center. HF: homogeneous flat crust; HT: homogeneous crust with topography; HeT: heterogeneous crust with topography.

Figures 9 and 10 compare the observed leveling data over the period 1982–1999 and the correspondent model results. The latter were obtained by further optimizing the source overpressure using the leveling data 1982–1999, while keeping the same source location and size from the joint inversion of EDM and leveling data from 1985–1999 (c.f. Section 2.3). In this case, we reach a good agreement between the models and uplift for all three crustal models, with residuals within the observation error (Figures 9b and 10). Some discrepancy is still observed for the benchmarks southeast of the caldera border, since the uplift in

this area is probably controlled by the deformation along the Sierra Nevada block [55]. In this case, the $\chi^2$ value is reduced by 36% when we add topography (from HF to HT) and by a further 10% when we also add the heterogeneities (46% total decrease $\chi^2$ in from HF to HeT). Observed values and numerical results for the leveling data over the period 1982–1999 are reported in Table S4, Supplementary Materials.

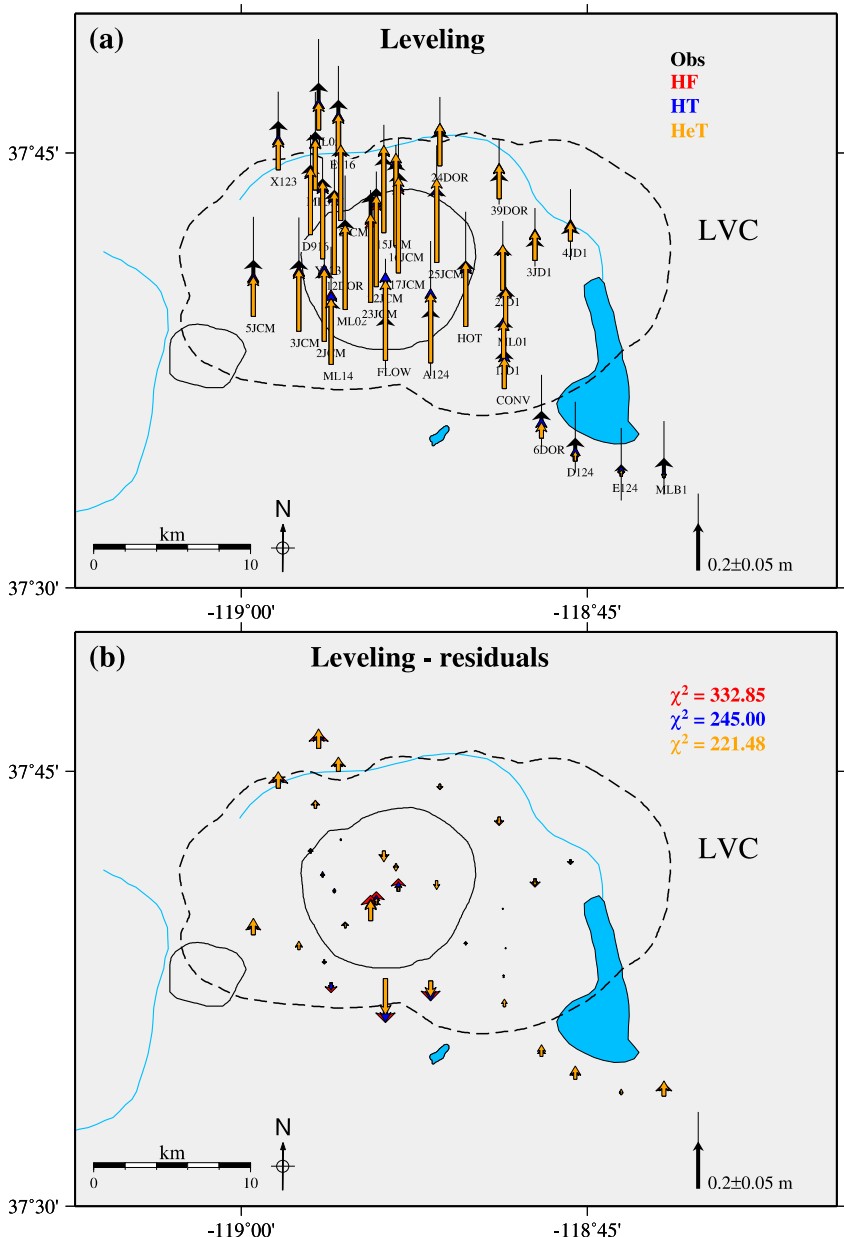

**Figure 9.** (**a**) Comparison between observed (black arrows with thin solid line as error) and modeled (colored arrows corresponding to different model configurations) uplift for the period 1982–1999. (**b**) Corresponding residuals between modeled and observed uplift. Obs: Observations; HF: homogeneous flat crust; HT: homogeneous crust with topography; HeT: heterogeneous crust with topography. $\chi^2$ values for each model are shown on top right.

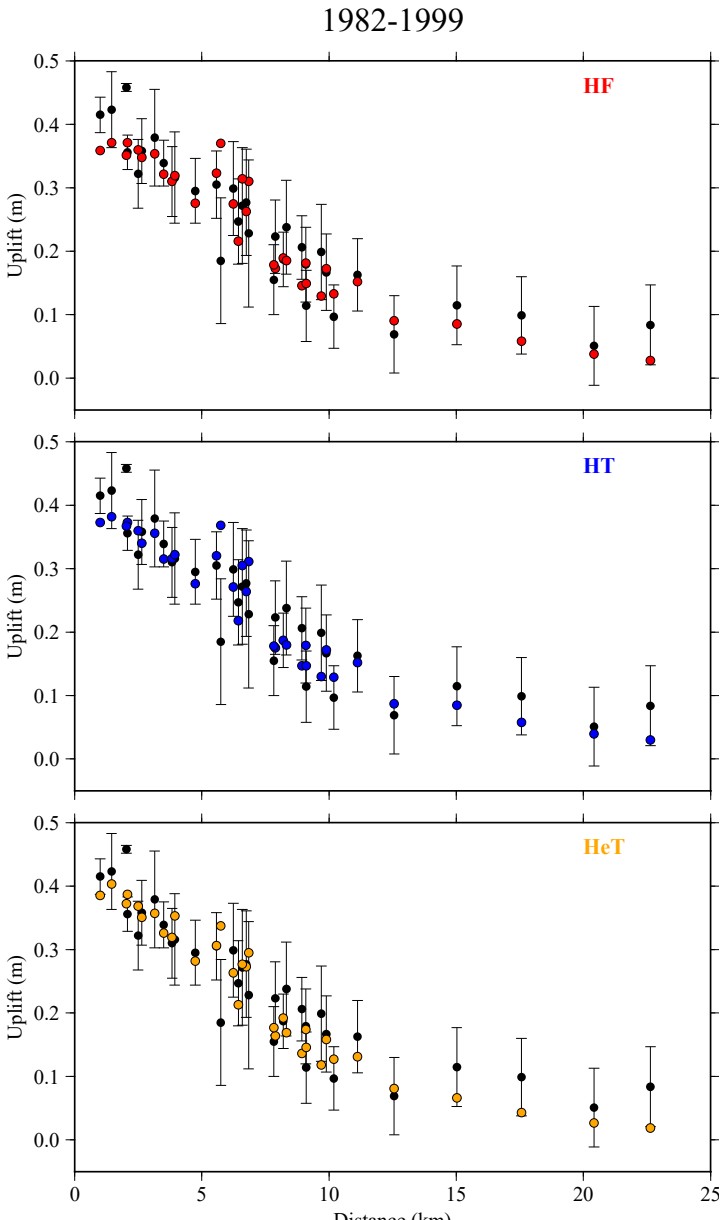

**Figure 10.** Comparison between observed (black circles with 1-sigma error bars) and modeled (colored circles corresponding to different model configurations) uplift at leveling benchmarks for the data in 1982–1999 ordered according to the horizontal distance from the resurgent dome center. HF: homogeneous flat crust; HT: homogeneous crust with topography; HeT: heterogeneous crust with topography.

The large residuals observed for three benchmarks close to the center of the resurgent dome are from the exploitation of the hydrothermal aquifers by the local geothermal power plant [13]. Other discrepancies are because of motion along faults in the caldera South Moat [4] or the Sierra Block (e.g., Figures 7 and 9) [8,55]. The heterogeneous model (HeT) can better fit the uplift for 1982–1999 than the two homogeneous models (HF and HT; Figure 9).

### 3.3. Density of the Intrusion

The observed gravity change, after free-air and water-table correction, shows a positive anomaly centered on the resurgent dome, with peak amplitude of about 60 μGal ([28] and

black circles in Figure 11), that suggests mass intrusion into the sub-caldera crust beneath the resurgent dome.

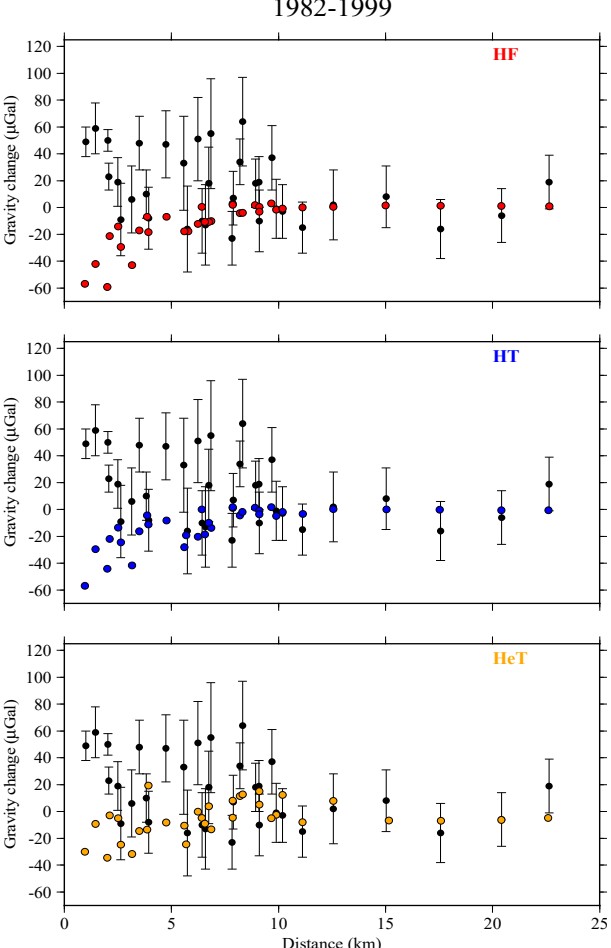

**Figure 11.** Comparison between observed gravity changes (1982–1999, Table S4, Supplementary Materials) after the removal of free air and water table effect (black circles with 1-sigma error bars) and the total deformation contribution to gravity changes ($\delta_{g1} + \delta_{g2\Delta V} + \delta_{g3}$) from the solution of Equations (7)–(9) corresponding to different model configurations (colored circles), ordered according to the horizontal distance from the resurgent dome center. HF: homogeneous crust, red circle; HT: homogeneous crust with topography, blue circle; HeT: heterogeneous crust with topography.

The estimate of the density of the intrusion requires three steps. First, we calculate the gravity changes associated with the deformation of the source and of the surrounding crust ($\delta_{g1} + \delta_{g2\Delta V} + \delta_{g3}$) by solving (7)–(9), the so-called "deformation effects", see Figure 11. The gravity variations due to deformation effects are substantial in the near-field of the source location, with the highest values at the source-tip location (up to $-60$ μGal, i.e., comparable, in magnitude, to the observed gravity change after free-air and water-table correction) and decaying to magnitudes <10 μGal at a distance of ~10 km.

We then subtract the deformation effects from the observed gravity changes in order to calculate the residual gravity (black circle with error bars in Figure 12). It is worth noting that the observed gravity changes had been previously corrected for water table noise and the free-air effect (Table S4, Supplementary Materials) (details in [28]). The residual gravity, $\delta_{g2V}$, depends on the mass change accompanying the deformation. Following the methodology described in Section 2.3, we solve the inverse problem with Poisson's Equation (10) to obtain the density value for the intruding fluid which best matches the observed residual gravity change $\delta_{g2V}$. Numerical results for modeled residual gravity change agree within the measurement errors with most of the residual gravity observations

(Figure 12). Figure 12 shows a good fit to the observed residual gravity for the FEM model of a homogeneous crust with topography (HT). Adding additional information about heterogeneities in the crust does not significantly improve the fit.

Table 4 shows the resulting density values of the intrusion, assuming that the mass change is due to either an incompressible ($\rho_{in}$) or compressible ($\rho_{cmp}$) fluid intrusion. The relation between the density of incompressible ($\rho_{in}$) or compressible ($\rho_{cmp}$) fluid intrusion is given in Equation (13). Reference [53] and Equation (12) allow for computing the compressibility $\beta_c$ of the crust surrounding the intrusion from quasi-static elastic properties (Table 1) and the results of the FEM models (Table 4).

The density of the intrusion depends on magma and reservoir compressibility— Equations (11)–(13). According to [56], the isothermal compressibility for a rhyolite, with a volume fraction of water between 0% and 30%, is $0.026 \pm 0.0011\,\text{GPa}^{-1}$. We estimated a crust compressibility of $0.147 \pm 0.037\,\text{GPa}^{-1}$ for the heterogeneous model. Using these values, we obtained a density of $1856 \pm 72\,\text{kg/m}^3$. This density is consistent with a rhyolite melt (no crystals) with 20% to 30% of dissolved hydrothermal fluids.

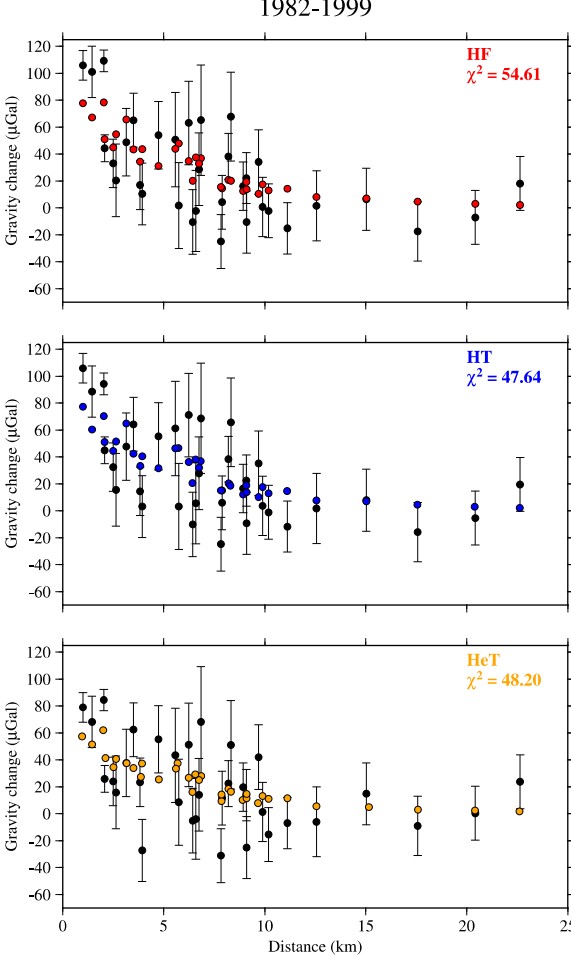

**Figure 12.** Comparison between observed (black circles with 1-sigma error bars) and modeled (colored circles corresponding to different model configurations) residual gravity, $\delta_{g2V}$. The misfit value $\chi^2$ for each case is also indicated. The corresponding best fit values for density, under the assumption of incompressible magma, are shown in Table 4. HF: homogeneous crust, red circle; HT: homogeneous crust with topography, blue circle; HeT: heterogeneous crust with topography, orange circle.

## 4. Summary

We build a 3D finite element model to investigate the source of observed displacements and gravity changes at Long Valley Caldera for 1982–1999. Using the geodetic data available in Long Valley Caldera GIS Database (https://doi.org/10.3133/ds81, accessed on 15 February 2021)—EDM baseline change, uplift, and gravity change measurements—we explore different model configurations starting from a flat and fully homogeneous domain and then adding additional complexities, such as topography and medium heterogeneities. Limits on the coverage of the existing geodetic monitoring network, ambiguities on the interpretation of subsurface distribution of the crust elastic properties, and the nature of non-linear inversion make our models and solutions non-unique. We can improve the bounds on the parameters of the deformation source by employing an appropriate modeling approach.

Since the joint inversion of horizontal and vertical deformation better constrain the geometry of the deformation source, we first invert EDM and leveling data for the period 1985–1999 to infer the size and location of an ellipsoidal source under the caldera. We then optimize the estimate of the source volume and mass change by performing a second set of inversions of leveling and gravity data for 1982–1999. This is the period with the best signal to noise ratio for gravity data.

One advantage of our approach is, not only the inclusion of topography, but also of full heterogeneities (3D). The influence of mechanical heterogeneities in LVC has been considered in other works [30,57–59], but while past works relied on simplified models in terms of geometry (e.g., 2D) or the material heterogeneities distribution (e.g., only vertical), in this study, similarly to [8], we implement both lateral and vertical material heterogeneities. Furthermore, we account for the difference between static and dynamic mechanical properties, since the use of a dynamic bulk modulus for the overall domain would overestimate the medium rigidity at shallow depths (lithostatic pressure < 3 kbar).

To estimate the subsurface mass change of the deformation source, we first estimate the so called "deformation effects" given by (7)–(9); see Figure 11. We then subtract the "deformation effects" from the gravity changes to obtain the residual gravity, since the residual gravity, $\delta_{g2V}$, depends on the mass change accompanying the deformation. Finally, we solve the inverse problem with Poisson's Equation (10) to obtain the density value for the intruding fluid which best matches the observed residual gravity change (Figure 12).

The density of the intrusion will change if the fluid is either incompressible ($\rho_{in}$) or compressible ($\rho_{cmp}$), since the density of the intrusion depends on the ratio $\beta_m/\beta_c$ (Equation (13), Table 4). We can estimate $\beta_c$ either from the shear modulus and source aspect ratio (Table 1; [53]) or from the absolute volume of the source inferred from our numerical analysis (Table 4). Both approaches calculate the same value of $\beta_c$ for a homogeneous medium. We find the second approach more appropriate for our heterogeneous model.

## 5. Conclusions

Gravity data are usually noisier than deformation data (e.g., [28]) but are essential for estimating the density of intrusion, because changes in the gravity potential are related to the changes in density distribution caused by the subsurface mass redistribution. Without gravity data, we cannot obtain information about the nature of the deformation. In this specific case, the major ambiguity is not coming from the errors in the gravity data but from the uncertainty about the appropriate value of magma compressibility. Reference [56] present experimental values of the isothermal compressibility of rhyolite, andesite, and basalt glasses as a function of the volume fraction of water (see Table 4 and Figure 5 in [56]). We assume here the compressibility values for rhyolite, i.e., the main component of erupted magma in LVC [18]. According to [56], the isothermal compressibility for a rhyolite, with a volume fraction of water between 0% and 30%, is $0.026 \pm 0.001$ GPa$^{-1}$. Using our inversion results for source parameters, for the heterogeneous case we estimated a reservoir compressibility of $0.147 \pm 0.037$ GPa$^{-1}$ (Table 4). We therefore obtained a

density of $1856 \pm 72$ kg/m$^3$. This density is consistent with a rhyolite melt with 20% to 30% of dissolved hydrothermal fluids.

**Supplementary Materials:** The following are available online at https://www.mdpi.com/article/10.3390/rs13204054/s1, Table S1: Horizontal deformation for the period 1985–1999 (two-color EDM data), Table S2: Vertical deformation (uplift) for the period 1985–1999, Table S3: Vertical deformation (uplift) for the period 1982–1999 (leveling data), Table S4: Gravity data (1982–1999).

**Author Contributions:** Conceptualization, F.P. and F.S.; methodology, F.P. and F.S.; software, M.B.; validation, F.P. and F.S.; formal analysis, F.P.; investigation, F.P. and F.S.; resources, M.B.; data curation, M.B.; writing—original draft preparation, F.P. and F.S.; writing—review and editing, all authors; visualization, M.B.; supervision, M.B.; project administration, M.B.; funding acquisition, M.B. All authors have read and agreed to the published version of the manuscript.

**Funding:** This research was funded by the—U.S. Geological Survey, Volcano Science Center, and by Sapienza—University of Rome, Piccoli Progetti Universitari 2020.

**Institutional Review Board Statement:** Not applicable.

**Informed Consent Statement:** Not applicable.

**Data Availability Statement:** Gravity and deformation data employed in this paper are available on-line from the Long Valley Caldera GIS Database ([13]; https://doi.org/10.3133/ds81, accessed on 15 February 2021), and as Supplementary Materials.

**Acknowledgments:** The authors wish to thank James Hickey (Bristol University) and Gilda Currenti (INGV) for the discussion about the model implementation. Mehdi Nikkoo (GFZ) for supporting the benchmark studies with analytical solutions and for the helpful conversations. The authors thank the Reviewers for the thorough and insightful comments which helped to improve the manuscript and Emily Montgomery-Brown (USGS) for internal review of the manuscript. Any use of trade, firm, or product names is for descriptive purposes only and does not imply endorsement by the U.S. Government.

**Conflicts of Interest:** The authors declare no conflict of interest.

## Appendix A  Validation of Numerical vs. Analytical Solution

To assess the accuracy of the finite element calculations, the absence of edge effects and ensure that the geometry and mesh adopted yield sufficient sensitivity, we validated the FEM, the numerical solution, for a vertical prolate ellipsoid in a homogeneous model without topography against the analytical solutions from [43]. For this comparison, we used a prolate ellipsoid with horizontal semiaxes a = b = 1.5 km and vertical semiaxis c = 3 km. The ellipsoid depth is 7 km and an outward (inflating) overpressure of 70 MPa is applied. We first compare the horizontal and vertical components of displacement on the stress-free surface on $\pm 40$ km EW profile (Figure A1a). Successively, we calculate the gravity changes due to the source deformation and to an intruding fluid with a density of 2700 kg/m$^3$ using the method described in the main text (Section 2.4) and compare them with the analytical solutions from [47] (Figure A1b,c). For the gravity contribution from the mass change ($\delta_{g2V}$), we also checked with the solutions from [60]. We observe a good agreement between the analytical and FEM solutions for both the displacements and the gravity components. Minor differences visible over the first 5 km distance from the source are inside the ranges of uncertainty, which are ~1 mm for horizontal displacements, ~1 cm for vertical displacements and ~10 µGal for gravity changes.

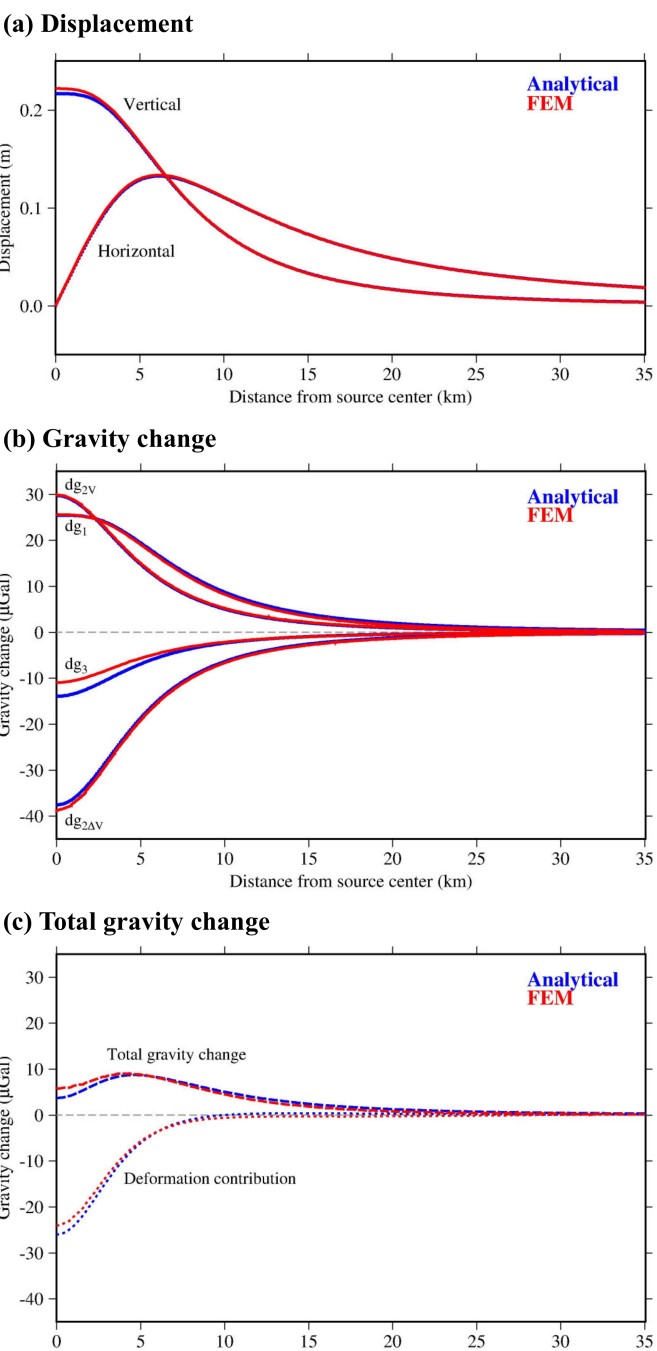

**Figure A1.** Benchmark test. Comparison between analytical (blue) and numerical (red) solutions for an ellipsoidal source in a homogeneous space with no topography. All solutions are shown along a radial distance centered at the source horizontal position. (**a**) Comparison of horizontal and vertical displacement at the free surface. (**b**) Comparison between the various components of gravity changes at the surface. (**c**) Total gravity change obtained by summing all the components shown in (**b**) and the gravity change only due to the contribution of deformation ($\delta_{g1} + \delta_{g2\Delta V} + \delta_{g3}$).

**Appendix B  Error Propagation**

$$A = \frac{E_c}{\sqrt{E_a E_b}} \quad \sigma_A^2 = 4\left[\frac{\sigma_{E_c}^2}{E_a E_b} + \frac{(E_b E_c)^2}{4(E_a E_b)^3}\sigma_{E_a}^2 + \frac{(E_a E_c)^2}{(E_a E_b)^3}\sigma_{E_b}^2\right] \tag{A1}$$

$$V = \frac{4}{3}\pi E_a E_b E_c$$
$$\sigma_V^2 = \frac{16}{9}\pi^2 \left[ (E_b E_c)^2 \sigma_{E_a}^2 + (E_a E_c)^2 \sigma_{E_b}^2 + (E_a E_b)^2 \sigma_{E_c}^2 \right] \tag{A2}$$

$$\Delta V \sim \frac{\Delta P}{\mu} V \quad \sigma_{\Delta V}^2 \sim \frac{1}{\mu^2} \left[ V^2 \sigma_{\Delta P}^2 + \Delta P^2 \sigma_V^2 \right] \tag{A3}$$

$$\beta_c = \frac{1}{V}\frac{\Delta V}{\Delta P} \quad \sigma_\beta^2 = \left[ \left(\frac{\Delta V}{V^2 \Delta P}\right)^2 \sigma_V^2 + \left(\frac{1}{V \Delta P}\right)^2 \sigma_{\Delta V}^2 + \left(\frac{\Delta V}{V \Delta P^2}\right)^2 \sigma_{\Delta P}^2 \right] \tag{A4}$$

$$r_v = 1 + \frac{\beta_m}{\beta_c} \quad \sigma_r^2 = \left[ \frac{\sigma_{\beta_m}^2}{\beta_c^2} + \frac{\beta_m^2}{\beta_c^4}\sigma_{\beta_c}^2 \right] \tag{A5}$$

$$\rho_{cmp} = \frac{\rho_{in}}{r_v} \quad \sigma_\rho = \frac{\rho_{in}}{r_v^2}\sigma_r \tag{A6}$$

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
