# Peer review of "A New Analysis of Caldera Unrest through the Integration of Geophysical Data and FEM Modeling: The Long Valley Caldera Case Study"

_remotesensing, doi:10.3390/rs13204054_

Round 1

Reviewer 1 Report

The study shows a finite element inversion of legacy EDM, leveling and gravity data of one of the best studied episodes of ground deformation at an active volcano - the 1980s-1990s episode of uplift at Long Valley caldera. A new analysis of legacy data with modern tools is always a good scientific contribution.  As I’m an InSAR and deformation specialist, not a gravity expert I will skip the details of that section and I will focus on the deformation data and modeling. 

One of the keypoints is that the source location and dimensions are nearly the same for all models, no matter how complex they are. This is something that should be better highlighted because it means that analytic models are still valid tools to study deformation at Long Valley.  However, the crustal heterogeneity results in lower values for the source overpressure compared to that of an homogeneous half-space elastic model as it has been shown in several studies  (e.g., Masterlark, 2007). Again, this is something that should be highlighted. 

There have been discussions for decades about the nature of deformation at Long Valley  (Hildreth, 2017; Hill et al., 2020), and the manuscript could do a better job bringing those scientific aspects to the manuscript's introduction. 

My major criticism comes from the gravity data interpretation. Even though I’m not an expert in this type of data, the conclusions drawn from their analysis are quite weak. This data set has a much larger scatter than the deformation data. This makes me wonder whether the gravity data is useful or not because their error bars are sometimes 50% of the actual measurements. Also, another major drawback is that the gravity and deformation data are inverted independently. Is it possible to jointly invert them? My guess is that it is not possible due to complications with the FEM setup, but I might be wrong. Finally, the authors could do a much better job explaining the implications for the nature of the fluids resulting in the deformation, as it has been done in previous studies  (Hildreth, 2017; Hill et al., 2020). In the current version it feels like they are not able to resolve between scenarios of a dying intrusion from which volatiles exsolve or magma injection. This could be an issue that it might not be possible to better constrain due to the uncertainties in the magma compressibility estimates in the absence of direct estimates in the rv ratio in equation 13. Also, the analysis of the magma compressibility is weak because they do not take into account cases when the injecting fluids are volatile rich. If the magma is compressed in the reservoir, then its pressure increases as it does the amount of H2O it can dissolve. 

Due to the potential issues with the gravity interpretation I recommend major revisions to the manuscript. In the following I have made comments that are easy to address.  When comments are within “” they are grammar corrections to the text. 

Abstract: it should include the main model parameters of the inverted source model (ie., spheroidal source at a depth of X km next to the resurgent dome). 

Line 19: typo “However, it is not clear”

Line 27: this is a rather general conclusion. Can you be more specific?

Line 34: I’d add that the caldera was formed  by the eruption of the Bishop tuff. I’d also include some references.

Line 35: cite this, for example figure 3 in Hill et al., 2020

Figure 1: include the SMSZ.

L61: “remains debated”

L60-63: long sentence, halve it.

L64: “insights”

L71-79: these are the general factors that must be addressed in microgravity studies. Since they are general, I’d remove the paragraph unless the authors state that all these factors are actually quite difficult to take into account. 

L85: “point source”

L89: “In this study we consider the 1982–1999 unrest period. This episode has the best and most complete gravity data set”

L109: Would InSAR from the ERS-2 mission provide some additional information ? For example Liu et al., 2011

L121-122: this is not well written.

L143-144: “Inside the domain, we assume the existence of an internally pressurized ellipsoidal prolate cavity that we invert for its location, size and overpressure” 

L148: “with the ellipsoidal source in its center”

L152: “, and  a fully heterogeneous”

L167: “Laboratory tests show” 

L172: I’d add a statement that this conversion from dynamic to static modulii is probably not unique, and just one of the several ways that these values can be converted into one other.

L186-187: “we prescribe a parametrized overpressure on the boundaries of the ellipsoidal cavity”

L190: “are carried until  an optimal performance” 

L191: “the mesh for the whole domain”

L204: “we infer the best-fit source geometry, position and overpressure”

L205. I’d state that the underlying assumption is that the deformation source is stable in location over time. 

L225: “oriented along the cartesian x, y and z axes respectively” 

L228: I’d also state that the source is assumed to be vertical with a plunge of 90º.

L231: I think that “search” is a better word than “constrain” in the context of the study because these non-linear inversion schemes carry out a smart search. 

L245-246: this was already said earlier in the manuscript. 

Section 2.4 I am not a specialist in gravity computations so I will skip comments form this section.  

L329-331: This is confusing. If the source for the HeT crust model is smaller, and located in the same place than the other sources, then the pressure change should be higher to predict the recorded displacements. Please clarify.

L 333: this figure is not necessary in the manuscript. Also it lacks scales so it is not clear how different are the starting models from the final ones.  It can be moved to the supplementary material. 

L338-340: this should be located right after line 331. Also, the reduction effects is not significant, from 69 to 65 Mpa which is ~5%.

L362: “MIKE is within the data uncertainty”

L380: “The inversion results for the leveling data (Fig. 7a) show”

L382-385: long sentence, halve it

L390: this point is not clear in Figure 2 due to the 3D view of the distribution of material properties.

L404: this seems to be secular deformation due to tectonic processes. Since the time scale of these processes is different to that of volcanic processes, are they negligible in the model?

L453-454: “the mass change is due to either an incompressible

Table 4. It seems that the gravity data do not have the resolving power to discriminate between an incompressible or compressible fluid intrusion. If so, this should be in the manuscript. Obviously this is not the author’s fault, just a characteristic of the data. 

L462-463: why? magma can be compressible if it is H2O-rich, but this is not addressed in the study.

L471-473: this should be stated  earlier in the manuscript when the authors introduce the rationale behind all their models.

L487: “inversion make our”

L503: “estimate the medium rigidity at lower depths”

L510-513: rewrite this sentence because the English is cumbersome. 

L531: “crystals and a very low”

L530: this is not correct. The magma can have a high viscosity in the absence of crystals if it is partially solidified as glass.

Reviewer 2 Report

The authors treated ground deformation data (EDM and leveling-GNSS) and gravity data and well analyzed by FEM modeling. Description of data and analysis is detailed and this manuscript is valuable to publish on Remote Sensing. 

I have just minor comments

Line 87: I cannot well understand the meaning of "hybrid-magma plus fluid intrusion".

Line 108: two color EDM?  

Figure 2: "d)" in the lower-left panel may be "e)", and "e)" on the lower-right may be "f)".

Equation (1)-(3) ν, ρ and E may be Poisson ration, density and Young modulus. It is better to write the meaning of these symbols in the text.

Table 1: It is better to describe the meaning of βc.

Line 157-158. Vs data [37] are used. Vp data are derived from Vp/Vs ratio. I think there are some research on Vp distribution beneath LVC. Why you did not use the Vp data which was obtained in the previous studies?

Figure 4: It is better to show scale.

Line 338: "P" may be "ΔP".

Line 341: Total displacement means combination of vertical and horizontal displacements?

Table 4: There are two columns for βm and rV, respectively. Do these columns correspond to two columns and  for βc ?

4. Discussion and conclusions: This chapter is composed of mostly summary of analysis and results. Interpretation part at the end follows Hildreth, W. (2017). Do you have any new insight for magma or fluid? I hope to add new discussion.

Reviewer 3 Report

Attached file
